# Passivation Effect of the Chlorinated Paraffin Added in the Cutting Fluid on the Surface Corrosion Resistance of the Stainless Steel

**DOI:** 10.3390/molecules28093648

**Published:** 2023-04-22

**Authors:** Lan Yan, Xingguo Yao, Tian Zhang, Feng Jiang, Yan Shui, Hong Xie, Zhiyang Xiang, Yousheng Li, Liangliang Lin

**Affiliations:** 1College of Mechanical Engineering and Automation, National Huaqiao University, Xiamen 361021, China; 2Institute of Manufacturing Engineering, National Huaqiao University, Xiamen 361021, China; 3Dongfang Turbine Co., Ltd., Deyang 618000, China; 4Xiamen Golden Egret Special Alloy Co., Ltd., Xiamen 361006, China

**Keywords:** stainless steels, cutting fluid, chlorinated paraffin, electrochemical characterization, corrosion resistance

## Abstract

Cutting fluids are the most effective method to lower the cutting temperature and decrease the cutting tool wear. At the same time, the cutting fluids influence the corrosion resistance property of the machined surface. In this study, chlorinated paraffin (CP), which is a common additive in the cutting fluid, was selected as the research objective to study its corrosion resistance property. The passivation effect of CP with different concentrations on the machined surface of stainless steel was studied. Electrochemical measurements and surface morphology investigation were used to characterize the passivation effect of CP with different concentrations. The test results showed that the corrosion resistance of stainless steel in the cutting fluid was enhanced with the increase in CP additive. This reason is that the charge transfer resistance increases and the corrosion current density decreases with the increase in CP additive. The X-ray photoelectron spectroscopy (XPS) results show that the proportion of metal oxides on the processed surface of the stainless steel sample was increased from 20.4% to 22.0%, 32.9%, 26.6%, and 31.1% after adding 1 mL, 2 mL, 4 mL and 6 mL CP in the cutting fluid with a total volume of 500 mL, respectively. The oxidation reaction between CP and the stainless steel sample resulted in an increase in metal oxides proportion, which prevented the stainless steel sample from corrosion in cutting fluid.

## 1. Introduction

Stainless steel, which contains 9–12 wt% chromium, is widely used in industrial [1] and biomedical [2,3] fields. Most stainless steels are difficult-to-cut materials due to high strength and low heat conductivity, which leads to high cutting temperature and fast cutting tool wear [4]. Cutting fluids are the most effective method to lower the cutting temperature and decrease the cutting tool wear [5]. Generally, cutting fluid has the main function of lubricating, cooling, and corrosion resistance. In ancient times, people have known that watering and olive oil can improve the quality and efficiency of finished products when processing metal parts and stone tools. In modern times, American scientist Taylor found that a heavy stream of water can improve the speed of the cutting tool by 30–40%, meanwhile, adding sodium carbonate to prevent rusting [6]. Since then, the discipline of cutting fluid has been developed. The focus of cutting fluid researching is mainly on tribological property and cooling functions. However, the risk of corrosion caused by additives of cutting fluid is usually ignored. There are only a few articles on corrosion caused by cutting fluid. Zhu J. et al. [7] studied corrosion behavior of Fe-based superalloy in cutting fluid. The results show significant corrosion of Fe-based superalloy after prolonged exposure to cutting fluid. Yan P. et al. [8] studied the effect of cutting fluid on stainless steel material. The results indicated that corrosion occurred related to the local loss of elemental chromium by using cutting fluid.

The workpiece and cutting tool in the environment suffer extremely high temperature and pressure [9], so chlorinated paraffin (CP) is widely used in the manufacturing industry as extreme pressure additives, due to its excellent tribological property. This excellent tribological property is attributed to the good chemical activity of CP, which leads to the formation of a metal-chlorinated film on the surface of workpiece. The metal-chlorinated film performs as a friction-reducing layer between the workpiece and cutting tool [10]. There is extensive literature that reported the importance and application of extreme pressure additives in the manufacturing process [11,12,13,14,15,16]. I. M. Petrushina et al. [17] briefly studied the chemical interaction of EP-additive (dialkylpoly sulfides and chlorinated paraffin) on AISI 304 stainless steel surface during ironing. It was found that chlorine additives have better extreme pressure performance in strip reduction tests, and it resulted from the better chemical activity of chlorinated paraffin with the main component (nickel, chromium, iron, etc.) of stainless steel. The study on the interaction of haloalkanes with metal has been conducted in the early years [18]. It has been reported that haloalkanes as acceptors of electrons, or oxidants, are generally reduced by iron. CP (40% Cl element containing) has better electrons acceptability than CP (70% Cl element containing) because chlorine atoms attached to adjacent carbon atoms are far apart from each other, which results in lower carbon–halogen bond energy. The reaction equation for the reduction of a haloalkane by iron to release the free halogen anion is shown below [19]:$$\left[-C-C-X\right]\overset{{\mathrm{Fe}}^{2+}}{\to }\left[-C-C-\right]+X^{-}$$

The oxidation properties of CP make it a potential anti-corrosion agent. As mentioned in literature [20,21,22,23], adequate dense oxide passivation film formed on the surface of stainless steel has been an effective way to prevent corrosion. There are no reports of CP applied in corrosion protection. This fact draws our attention to the oxidation reaction of CP between stainless steel workpieces. Thus, it is important to study the passivation mechanism of CP with a stainless steel workpiece, which helps improve the machining process of stainless steel.

At present, electrochemical measurements are the primary method for the research of corrosion and passivation. In the electrochemical measurements, the relationship between the given voltage and the output current helps us understand corrosion or passivation behavior indirectly [24,25,26,27,28]. The potentiodynamic polarization measurement, or Tafel polarization method, is widely applied to the corrosion analysis field. It is kinetics based on the dissolution of anodic metal that allows faster electrochemical measurements to predict the long-period corrosion [29]. The electrochemical impedance spectroscopy (EIS) measurement is a kind of non-destructive testing technology. It is a method to obtain the relationship between electrode polarization dynamics and electrode surface state by applying different frequencies of the sinusoidal current to the electrode surface. It analyzes the separate layers or structures on the electrode surface by creating an equivalent circuit [30,31].

In this study, the passivation behavior of stainless steel in the cutting fluid with chlorinated paraffin (CP) of different content was studied using several characterization methods. The potentiodynamic polarization and electrochemical impedance spectroscopy (EIS) were selected for obtaining corrosion current density and charge transfer resistance, respectively. Scanning electron microscope (SEM) and energy disperse spectroscopy (EDS) were selected for the surface morphology observation and the characterization of the elemental composition of the passivation film, respectively. X-ray photoelectron spectroscopy (XPS) was selected for the characterization of the reaction-generated products.

## 2. Results and Discussion

### 2.1. Electrochemical Characterization

#### 2.1.1. Potentiodynamic Polarization Curves of Stainless Steel in the Cutting Fluid

The potentiodynamic polarization measurement was used to measure the corrosion sensitivity of the stainless steel sample with CP of different content added in the E206 cutting fluid. In this study, *I*_corr_ reflects the corrosion rate directly. The electrochemical parameter *I*_corr_ is the ratio of corrosion current and contact area. The corrosion current and corrosion potential are determined in the intersection point of anodic polarization curve and cathodic polarization curve. A higher *I*_corr_ value means a higher corrosion rate. The potentiodynamic polarization curves obtained in deionized water and E206 cutting fluid in the absence and presence of different concentrations of CP are shown in Figure 1. The corrosion-related parameters included corrosion current density *I*_corr_ (μA cm^−2^), corrosion potential *E*_corr_ (V), and Tafel slope, which are shown in Table 1. Corrosion-related parameters were significantly influenced by the presence and different concentrations of CP in the E206 cutting fluid. Polarization curves shifted toward the negative direction with the addition of CP, as shown in Figure 1. As listed in Table 1, all *I*_corr_ values of CP-added groups were lower than E4 blank group.

It was found that the *I*_corr_ value of the stainless steel sample in deionized water was considerably less than that in the cutting fluid environment. It meant the utilization of cutting fluid indeed posed a risk of corrosion. All the *I*_corr_ values obtained from groups with the addition of CP were lower than that obtained from the E4 blank group, so the corrosion resistance of stainless steel in E206 cutting fluid could be effectively improved by adding CP.

The final potential of polarization measurement was increased from −0.6 V to 3.0 V and the complete anodic polarization curve is shown in Figure 2. It can be observed that the first peak appeared at −0.2 V, which is associated with the transition of the dissolution state and the pseudo passivation state. The pseudo passivation zone lasted for a short period in the range between −0.2 V and 0.3 V. The stainless steel sample continued to dissolute till the second peak corresponded to primary passive potential (*E*_pp_). It can be observed that a significant drop of current density appeared in the interval from 1.1 V to 2.5 V with the addition of CP, so a significant passivation zone appeared (range d–e in Figure 2). However, no obvious passivation zone appeared in the E4 blank group. In the primary passivation region, the current density lowers and almost stays constant. The value of *E*_pp_ depended on the oxidation of Cr to Cr^3+^ [32]. Group E4-CP2 (*E*_pp_ of 1.1 V) and E4-CP4 (*E*_pp_ of 1.16 V) reached the *E*_pp_ at almost the same potential. The *E*_pp_ of group E4-CP6 appeared at 1.25 V, and group E4-CP1 reached the biggest *E*_pp_ of 1.44 V. The most-generated product under this condition is dense film of Cr_2_O_3_, which protects the stainless steel sample from corrosion.

Then, the current density continued to increase as the potential increased and reached the transpassive state (range e–f in Figure 2). Then, the third peak appeared at approximately 2.5 V, known as secondary passive potential (*E*_sp_), which depended on the oxidation from Cr^3+^ to Cr^6+^ [32]. The most-generated product under this condition was CrO_3_. Groups E4-CP1, E4-CP2, E4-CP4, and E4-CP6 reached the *E*_sp_ at 2.46 V, 2.35 V, 2.5 V, and 2.46, respectively. The secondary passive state of anode electrode in alkaline solutions was consistent with literature [33,34]. The primary passive region area (Δ*E* = *E*_sp_ − *E*_pp_) was 1.02 V, 1.25 V, 1.34 V, and 1.21 V when 1 mL, 2 mL, 4 mL, and 6 mL CP was added, respectively. A larger passivation zone means better corrosion resistance. This result indicated that CP promoted the oxidation reaction on the surface of stainless steel sample. CP promoted the generation of passivation film on the surface of the stainless steel sample. The main composition of this passivation film needs to be further characterized using elemental analysis methods, such as EDS and XPS.

#### 2.1.2. Electrochemical Impedance Spectroscopy (EIS) Analysis of Passivation Film

The electrochemical impedance spectroscopy (EIS) measurements were performed after the polarization tests. It was utilized to obtain the passivation film information of the sample surface. The experiments and fitting results of the Nyquist plot are shown in Figure 3 and Table 2. As shown in Figure 3, all curves were in a similar shape and consist of two semicircles at the high frequency and medium frequency, respectively.

The Bode plot contains the phase angle-frequency plot, and the module of impedance-frequency plot is shown in Figure 4. This is consistent with the Bode-phase plot which means two different processes occurred. In the range of the second semicircle in the medium frequency region, the diameter of the semicircle increased with the addition of 1 mL CP and 2 mL CP, respectively. Then, it decreased slightly with the addition of 4 mL CP, and it increased again with the addition of 6 mL CP. It can be noted in Figure 4 that the phase angle and module of impedance increased as CP was added. The fitting curves, no matter the Nyquist plot or the Bode plot, showed good consistency with experiment results which reflected in the small relative error of those EIS parameters shown in Table 2. It contains the solution resistance (*R*_s_), the outer layer constant phase element (CPE1), the passivating film pseudo resistance (*R*_film_) [35,36,37], the inner layer constant phase element (CPE2), and the charge transfer resistance (*R*_ct_). All the above parameters were fitted by Zview software (Version 3.1, Scribner Associates Inc., Southern Pines, NC, USA).

The equivalent circuit was proposed as shown in Figure 5. As listed in Table 2, the solution resistance *R*_s_ in each set of experiments were around 35 Ω·cm^−2^, resulting from an electrolyte environment with similar conductivity as shown in Figure 6. The addition of CP caused a slight reduction in conductivity of cutting fluid relevant to the insulation of CP, and the mean of conductivity of those cutting fluids was 1.406 ms/cm. It is difficult to obtain a pure capacitor in the electrochemical process. Usually, CPE is used to approximately replace the capacitance. The CPE element is represented by Equation (1) [38,39,40]:(1)$$Z_{CPE_{dl}}=1/\left[T{\left(wj\right)}^{n}\right]\;(0<n<1)$$
where *T* is the imaginary admittance of the CPE, where w is the angular frequency, where $j^{2}=-1$, where *n* gets closer to 1, the more it tends to be a pure capacitor [41]. As listed in Table 2, most of the outer layer CPE1 values in the presence of CP were bigger than that of the E4 blank group. Meanwhile, the inner layer CPE2 values in the presence of CP were smaller than that of the E4 blank group. The CPE value is relevant to the thickness of the passivation film. When the dielectric constant is almost constant, the increase in CPE is caused by the decrease in the thickness of the passivation film [42]. It can be explained that the outer layer of the passivation film became thinner with the addition of CP. This was also reflected in the change of *R*_film_ that most of the *R*_film_ values decreased with the addition of CP. It was relevant to the formation of low-impedance porous metal hydroxides in the outer layer. The inner layer of the passivation film became thicker with the addition of CP, which was relevant to the generation of high-impedance and dense metal oxides film on the stainless steel sample surface. This was also reflected in the change of passive film resistance *R*_ct_. *R*_ct_ is relevant to the corrosion rate directly. A higher *R*_ct_ means a smaller corrosion rate. *R*_ct_ is inversely proportional to *I*_corr_ [43]. As listed in Table 2, all the *R*_ct_ values in the presence of CP were greater than the E4 blank group.

In summary, after adding CP, the values of CPE1 were slightly larger and the values of R_film_ were slightly smaller than the E4 blank group. It is relevant to the thinning of the outer corrosion film on the stainless steel sample surface. All the values of CPE2 were smaller than the E4 blank group and all the values of *R*_ct_ were larger than the E4 blank group. It is relevant to the formation of the thicker inner passivation film on the stainless steel sample surface. It can be speculated that the outer layer CPE1 tends to increase, while the pseudo resistance *R*_film_ tends to decrease, which is attributed to the formation of porous metal hydroxides in the outer layer. The inner layer CPE2 tends to decrease, while the charge transfer resistance *R*_ct_ tends to increase, which is attributed to the formation of dense metal oxides in the inner layer. The results of EIS measurements show good consistency with the results of potentiodynamic polarization measurements.

### 2.2. Observation of Surface Morphology and Component Analysis of Passivation Film

#### 2.2.1. SEM EDS Analysis of the Passivation Film

Four typical surface structures in this work are presented in Figure 7a–d. They are the original polished stainless steel sample surface, E4 blank group with a lot of pits, E4-CP2 group with rhombohedral pillar shape crystal, and E4-CP4 group with large number of surface defects, respectively.

It can be seen from these SEM photos that the original polished surface is relatively smooth, with only minor scratches after polishing with #800, #1000, #2000, and #4000 sandpaper in sequence. The experimental error caused by variation of surface roughness was reduced effectively, which is helpful for the implementation of electrochemical measurements. The EDS analysis of the stainless steel sample surface was carried out to identify the elemental composition of the different regions. As shown in Figure 7a, Point 1 is in the main body of the stainless steel sample. The results showed that the stainless steel sample is mainly composed of iron, accounting for more than 70% of the weight percentage. The content is followed by chromium, accounting for about 10% of the weight percentage. The rest of the elements consist of small amounts of carbon, manganese, cobalt, and other elements. Point 2 is in the highlighted part in the figure. It found that the main element in this region is tungsten, whose content has increased to about 50%. The EDS results of two different areas revealed that the main body of stainless steel sample consists of iron, chromium, and another small amount of elements such as tungsten, carbon, manganese, and cobalt.

As shown in Figure 7b, small pits appeared on the surface of the stainless steel sample. Oxygen content increased slightly and iron content decreased slightly after the electrochemical test. It can be observed in Figure 7c that a large area of defect appeared on the surface of the stainless steel sample. In addition, crystals in a rhombohedral pillar shape appeared inside the defects, which are associated with the formation of metal hydroxides by hydrolysis reaction of the metal surface with cutting fluid. It can be seen from the EDS spectrum of Point 5 and Point 6 that the oxygen content increased sharply. Additionally, a small signal of Cl element was detected, which is attributed to the generation of metal chloride and residues of CP. The concentration of Cl element is too low to penetrate into the passivation film, and it is the explanation why the electrochemical test results are inconsistent with the extensive literature that stainless steel is prone to corrosion in environments containing chloride ions. As shown in Figure 7d, the stainless steel sample surface has a large number of surface defects. It can be seen from the EDS spectrum of Point 7 and Point 8 that the main component of the highlight block is tungsten, and the content of oxygen also increased.

Through the SEM and EDS analysis results of the stainless steel sample surface, it was found that a large area of pits and defects appeared on the surface of the stainless steel sample after the electrochemical test. It was attributed to the forced loss of electrons on the stainless steel sample surface during the strong polarization process (range a–b) in Figure 2. The oxygen content on the surface of stainless steel sample increased significantly. It was attributed to the oxidation property of CP. The metal oxides content increased in the stainless steel sample surface as CP was added. The passivation film prevents the stainless steel sample from corrosion in cutting fluid. These results confirmed the hypothesis derived from the electrochemical tests.

#### 2.2.2. XPS Analysis of the Products in Corrosion Scale

The composition of the corrosion scale on the stainless steel sample surface was analyzed by XPS. The signal of Fe 2p, Cr 2p, O 1s, and Cl 2p was used to discuss the composition of the corrosion scale. To ensure the reliability of results, the high-resolution spectra of each element were fitted in exactly the same way [44]. A similar binding energy (BE/eV) and the same full width half maximum (FWHM /eV) were used. The high-resolution spectra of Fe 2p and Cr 2p are shown in Figure 8 and the high-resolution spectra of O 1s and Cl 2p are shown in Figure 9.

The fitting results including binding energy (BE/eV), full width half maximum (FWHM/eV), and area of each peak are shown in Table 3. The binding energy of each species was referred to in literature [44]. The composition of corrosion scale in all sets of experiments is basically the same. Fe 2p was decomposed into eight peaks, representing six different Fe species and two satellite peaks of Fe. These six kinds of Fe were iron (707 eV), FeO (709.4 eV), Fe_3_O_4_ (710.4 eV), Fe_2_O_3_ (710.8 eV), FeCl_3_ (711.3 eV), and α-FeOOH (711.8 eV), respectively. Fe 2p exhibited two different valence states of +2 and +3, while α-FeOOH owned the largest peak area among all iron species in each group. This result is consistent with the SEM image shown in Figure 7c. α-FeOOH was in rhombohedral pillar shape, which is consistent with the literature [45]. Cr 2p was decomposed into three peaks, representing Cr_2_O_3_ (576.8 eV), Cr(OH)_3_ (577.3 eV), and CrO_3_ (578.3 eV), respectively. The dense chromium oxide protected stainless steel from corrosion. Cl 2p was decomposed into two kinds of peaks attributed to organic chloride (200 eV) and metal chloride (198–199 eV). The signal of Cl 2p was relatively small, which is consistent with the EDS results, as shown in Figure 7c, and it is mainly composed of a small amount of residual cutting fluid and a small amount of metal chloride.

The change of O 1s spectrum showed that CP promoted the oxidation of the stainless steel sample. O 1s was decomposed into three peaks, attributed to metal oxides (529–530 eV), metal hydroxides (530–532 eV), and bounded water (532–534 eV) [46,47]. The peak area of metal hydroxides in all groups was the largest, which was attributed to the hydrolysis of metal in aqueous solution. It could be found that the peak area attributed to metal oxides increased with the addition of chlorinated paraffin. According to the quantification calculation, the proportion of metal oxides in groups E4, E4-CP1, E4-CP2, E4-CP4, and E4-CP6 was 20.4%, 22.0%, 32.9%, 26.6%, and 31.1%, respectively.

The formulation of XPS quantification calculation is presented by Equations (2)–(4) [48]:(2)$$A_{m}^{\prime }=A_{m}/SF_{m}$$
(3)$$A_{total}=A_{Fe}^{\prime }+A_{Cr}^{\prime }+A_{O}^{\prime }+A_{Cl}^{\prime }$$
(4)$$M\%=A_{m}^{\prime }/A_{total}$$
where $A_{m}$ is the peak area of species m,$\;SF_{m}$ is the sensitivity factor of species m, and $A_{m}^{\prime }$ is the normalized peak area of species m. The value of the sensitivity factor depends on the test instrument. In this study, the sensitivity factors of Fe 2p, Cr 2p, O 1s, Cl 2p3, and Cl 2p1 are 10.82, 7.69, 2.93, 2.285, and 0.961, respectively.

Chlorinated paraffin promotes the generation of metal oxidation corrosion scale, which contributed to the better corrosion resistance of stainless steel sample. Chlorinated paraffin is prone to REDOX reactions with metal in the stainless steel sample surface where iron and chromium are oxidized. According to the reaction below:$$R-{\mathrm{CH}}_{2}{\mathrm{CH}}_{2}\mathrm{Cl}+\mathrm{Me}+{\mathrm{ne}}^{-}\to R-\mathrm{CH}={\mathrm{CH}}_{2}+{\mathrm{HCl}}_{\left(\mathrm{aq}\right)}+{\mathrm{Me}}^{n+}$$

Initially, the metal ions undergo a hydrolysis reaction to form metal hydroxides according to the reaction:$$\mathrm{Fe}{\left(\mathrm{OH}\right)}^{+}+{\mathrm{OH}}^{-}\to \mathrm{Fe}{\left(\mathrm{OH}\right)}_{2}$$
$$4\mathrm{Fe}{\left(\mathrm{OH}\right)}_{2}+O_{2}+2H_{2}O\to 4\mathrm{FeOOH}\cdot H_{2}O$$
$$\mathrm{Cr}{\left(\mathrm{OH}\right)}^{+}+{\mathrm{OH}}^{-}\to \mathrm{Cr}{\left(\mathrm{OH}\right)}_{2}$$
$$\mathrm{Cr}{\left(\mathrm{OH}\right)}_{2}+H_{2}O\to \mathrm{Cr}{\left(\mathrm{OH}\right)}_{3}+H^{+}+e^{-}$$

Then, the part of the metal hydroxide continues to oxidize to form metal oxides according to the reaction:$$2\mathrm{FeOOH}\cdot H_{2}O\to {\mathrm{Fe}}_{2}O_{3}+3H_{2}O$$
$$2\mathrm{Cr}{\left(\mathrm{OH}\right)}_{3}\to {\mathrm{Cr}}_{2}O_{3}+3H_{2}O$$

Meanwhile, part of the metal hydroxides and metal oxides react with chloride ions to form metal chloride according to the reaction:$$\mathrm{FeOOH}+3\mathrm{HCl}\to {\mathrm{FeCl}}_{3}+3H_{2}O$$
$${\mathrm{Fe}}_{2}O_{3}+6\mathrm{HCl}\to 2{\mathrm{FeCl}}_{3}+3H_{2}O$$

In summary, CP can promote the surface oxidation of the stainless steel sample. The outer layer of the corrosion scale was mainly composed of porous metal hydroxides such as α-FeOOH and Cr(OH)_3_. The inner layer of the corrosion scale was mainly composed of dense metal oxides such as Cr_2_O_3_ and CrO_3_. These results are in agreement with the explanation of equivalent circuits to the corrosion scale. Corrosion resistance of the stainless steel sample in the environment containing CP is better than that in the pure cutting fluid environment, which is attributed to the presence of dense metal oxide corrosion scale.

## 3. Experiment Design

### 3.1. Preparation of Stainless Steel Sample for Electrochemical Characterization

The material selected for the research is a kind of hard-to-machine stainless steel, usually used in the manufacturing of turbine blades. The main component of stainless steel is shown in Table 4. The steel was machined into a cuboid by wire cut electric discharge machine with a size of 10 mm × 10 mm × 2 mm, then welded with copper wire. Epoxy resin was utilized to encapsulate it into a cylinder, and only one side was left as the working electrode for the convenience of the electrochemical measurements.

### 3.2. Preparation of Cutting Fluid and Electrochemical Characterization

In this study, commercial water-based cutting fluid E206 from Master Chemical Co., Ltd. (Plattsburgh, NY, USA) and CP (40% Cl element containing) were selected for the research of passivation behavior. The total volume of prepared cutting fluid is 500 mL for each group. The volume of E206 stock solution is a constant 20 mL, expressed in the volume fraction of 4%, which group is abbreviated as E4. In addition, chlorinated paraffin of 1 mL, 2 mL, 4 mL, and 6 mL were added in the group of E4, which are abbreviated as E4-CP1, E4-CP2, E4-CP4, and E4-CP6, respectively. The pH of the solution is between 8 and 9. Another blank group of deionized water was set. All the glassware used in this experiment was washed with ethanol and deionized water. All the cutting fluid was prepared freshly with deionized water before the experiment. A conventional three-electrode electrochemical cell was selected for this study. It contains a working electrode (WE) mentioned, a platinum foil counter electrode (CE) with an area of 1 cm^2^, and a Ag/AgCl reference electrode (RE), which is fully immersed in KCl solution (3 mol/L). The working electrode was encapsulated with epoxy resin. The working surface of the stainless steel was polished to the average surface roughness Ra of lower than 100 nm, which is measured by Zygo 9300 (Zygo Corp., Middlefield, CT, USA). The stainless steel samples were processed by ultrasonic cleaning in ethanol for 5 min and then washed with deionized water before each experiment. The platinum has little polarization in the electrochemical process. To prevent the pollution of the reference electrode by cutting fluid, which may lead to a slight change of reference potential during the experiments, a salt bridge containing 3 mol/L KCl solution was attached to the reference electrode. The tip of the reference electrode and the salt bridge are a porous ceramic junction that allows a low flow rate of cutting fluid to prevent pollution of reference potential by the cutting fluid and prevent pollution of cutting fluid by the reference potential.

All the electrodes were immersed in the cutting fluid for 30 min before electrochemical tests to obtain a stable open circuit potential (OCP), which is the potential between the reference potential and the working electrode. However, there is no current flow in the above period. All the electrochemical tests were performed under the stable OCP. The potentiodynamic polarization measurement was performed immediately. There is a current flow between the working electrode and the counter electrode, and the direction of the current changes in the process, which is called polarization current. Parameters such as corrosion potential (*E*_corr_), corrosion current (*I*_corr_), and Tafel slope can be obtained by the potentiodynamic polarization curve which reflects the corrosion rate directly. The range of potential was set as follows: the initial potential from −0.8 V to the final potential 0.6 V, and the scan rate to 1 mV/s. In addition, the final potential of another group was increased to 3.0 V to obtain the complete anodic polarization curve which was used to investigate the chemical process on the sample surface before and after adding CP. The electrochemical impedance spectroscopy (EIS) measurement was carried out after the polarization measurement attempt to obtain the information of the passivation film. The sinusoidal electrical signal with an amplitude of 5 mV and frequency from 10^6^ Hz to 1 Hz was applied to the working electrode under the open circuit potential to obtain the passive film information of the working electrode surface. All the above electrochemical tests were performed by CHI 660E electrochemical workstation (CHI Instruments, Shanghai, China). The electrochemical workstation was stabled for 30 min before electrochemical measurements.

### 3.3. Surface Morphology Observation and Passivation Products Characterization

After finishing the electrochemical tests, the stainless steel sample was ultrasonically cleaned by alcohol and deionized water for 10 min, respectively. The SEM and the EDS, provided by Phenom ProX (Phenom Word Corp., Eindhoven, The Netherlands), were performed for the observation of surface morphology and the element analysis of surface film products. The EDS analysis requires high energy, which contributed to electron beam energy set to 15 kV, and the analysis mode is point mode. EDS analysis was conducted in and around the reaction-generated product target, respectively. The XPS, provided by Thermo Fisher Scientific (Waltham, MA, USA) with an Al Kα source, was performed for the analysis of reaction-generated products. The reaction-generated products can be identified by distinguishing the binding energies of different substances. In this study, the element variation of Fe, Cr, O, and Cl on the processed surface was investigated. The results of the EDS and the XPS analysis are the key information for the explanation of the passivation mechanism.

## 4. Conclusions

In this study, passivation behavior of the stainless steel in the E206 cutting fluid with the addition of chlorinated paraffin has been investigated by electrochemical measurements and elemental analysis methods. It was found that the corrosion process of stainless steel in the E206 cutting fluid was alleviated by adding CP. It is related to the formation of a metal oxide layer on the electrode surface. There is a good correspondence between the results of electrochemical measurements and elemental analysis. The corrosion resistance of the stainless steel sample in cutting fluid can be enhanced as CP is added. The corrosion resistance was primarily attributed to the electronegativity of halogen substituents of CP, which result in the formation of dense metal oxides as the passivation film.

## Figures and Tables

**Figure 1 molecules-28-03648-f001:**
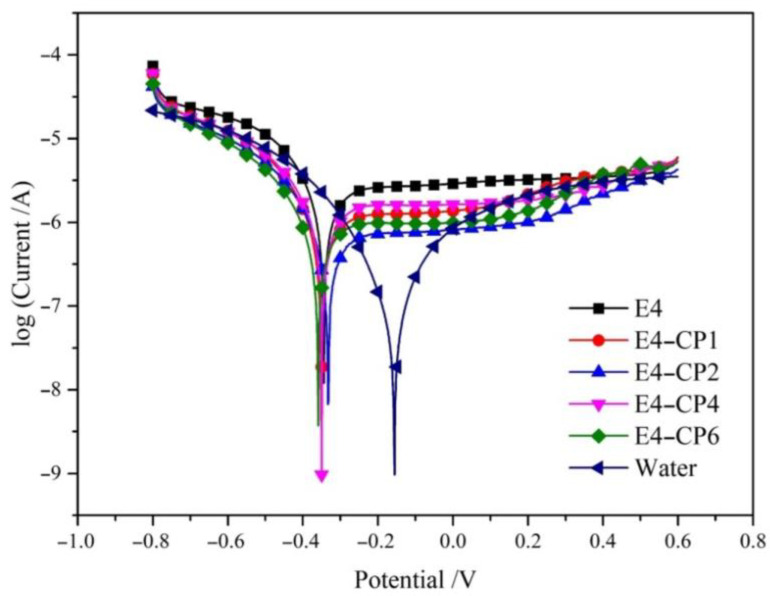
Potentiodynamic polarization curves of stainless steel in E206 cutting fluid added with CP after 30 min immersed under the stable OCP.

**Figure 2 molecules-28-03648-f002:**
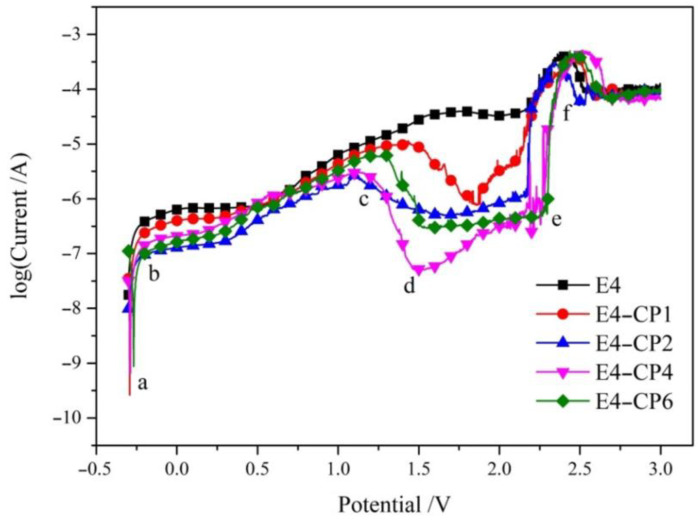
The complete anodic polarization curves with the final potential of 3.0 V obtained from group E4, E4-CP1, E4-CP2, E4-CP4, and E4-CP6, respectively.

**Figure 3 molecules-28-03648-f003:**
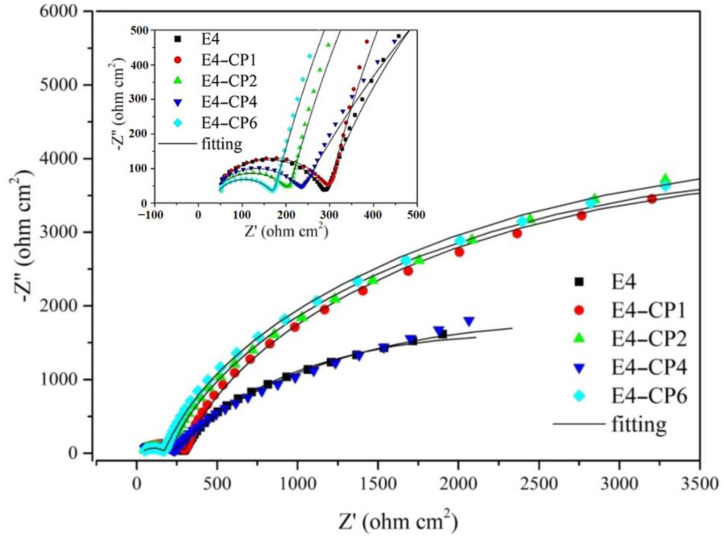
The Nyquist plot and fitting curves for passivation film formed after polarization measurements (frequency range from 10^6^ Hz to 1 Hz under the OCP; 25 °C).

**Figure 4 molecules-28-03648-f004:**
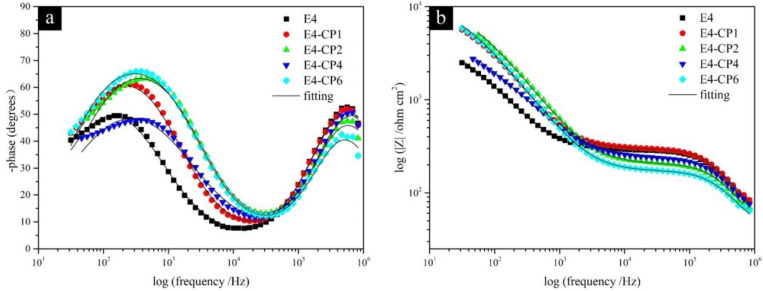
The (**a**) Bode phase plot and (**b**) Bode phase angle module of impedance and fitting curves for passivation film formed after polarization measurements (frequency range from 10^6^ Hz to 1 Hz under the OCP; 25 °C).

**Figure 5 molecules-28-03648-f005:**
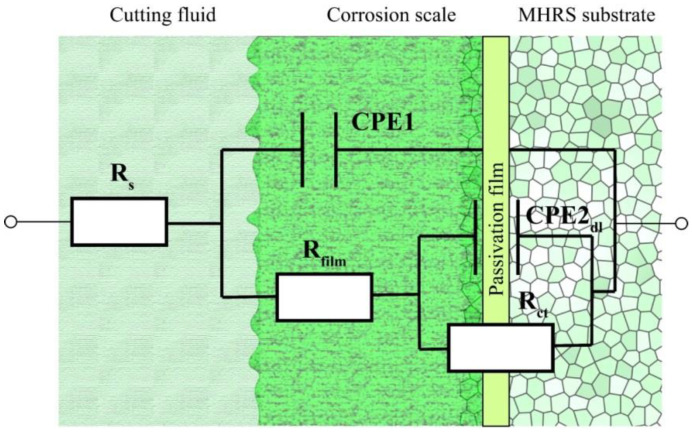
The equivalent circuit contains solution resistance (*R*_s_), outer layer constant phase element (CPE1), corrosion scale resistance (*R*_film_), inner layer constant phase element (CPE2), passive film resistance (*R*_ct_).

**Figure 6 molecules-28-03648-f006:**
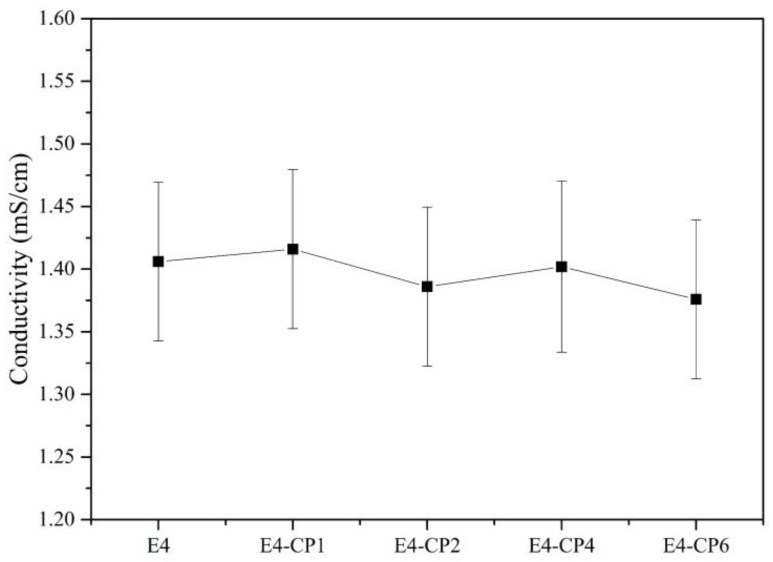
Conductivity of cutting fluid at different CP concentrations.

**Figure 7 molecules-28-03648-f007:**
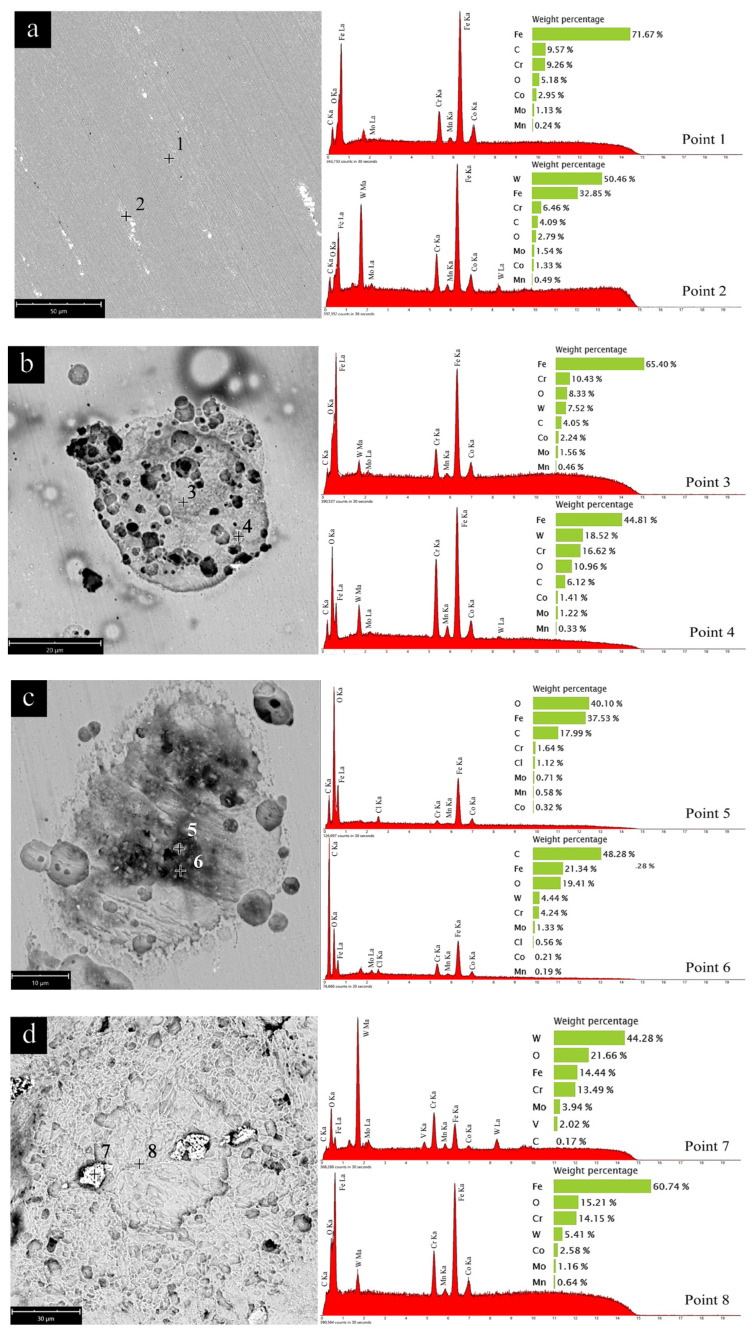
The scanning electron microscope (SEM), energy disperse spectroscopy (EDS) analysis of (**a**) the original polished stainless steel sample surface, (**b**) E4 group in pits shape, (**c**) E4-CP2 group in rhombohedral pillar shape, (**d**) E4-CP4 group with large number of surface defects.

**Figure 8 molecules-28-03648-f008:**
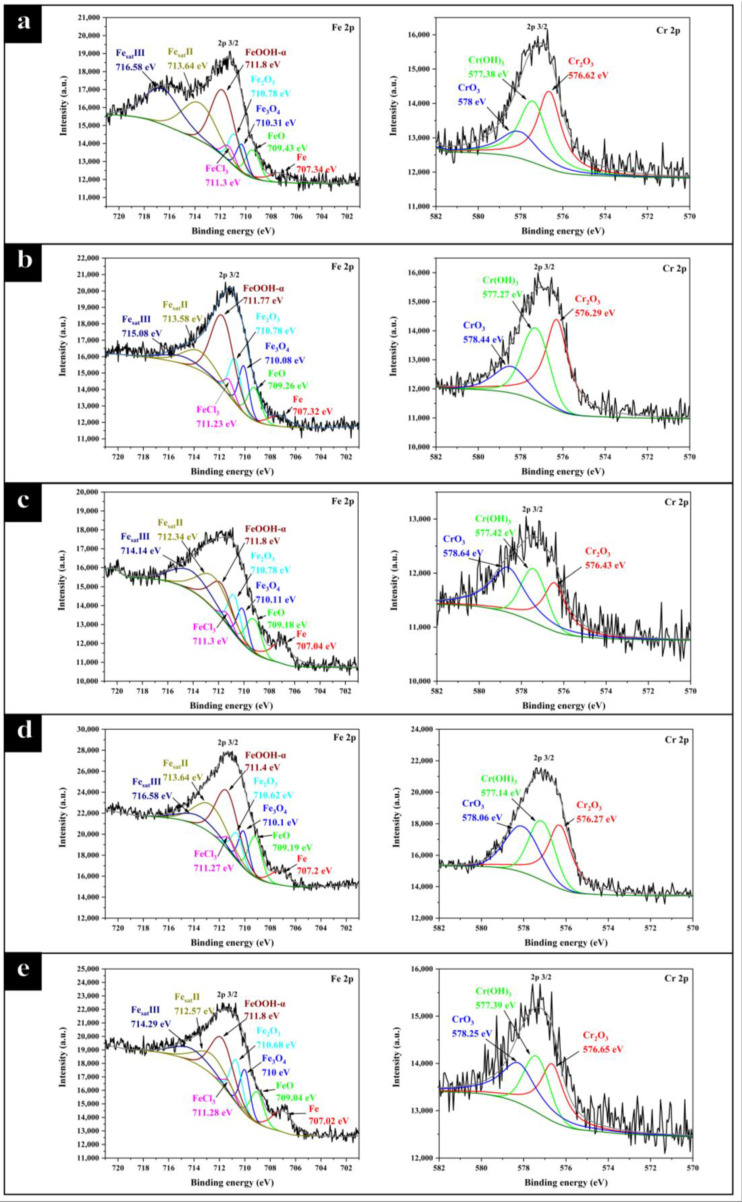
The XPS high-resolution spectrum with the signal of Fe 2p, Cr 2p (**a**) E4 group, (**b**) E4-CP1 group, (**c**) E4-CP2 group, (**d**) E4-CP4 group, and (**e**) E4-CP6 group.

**Figure 9 molecules-28-03648-f009:**
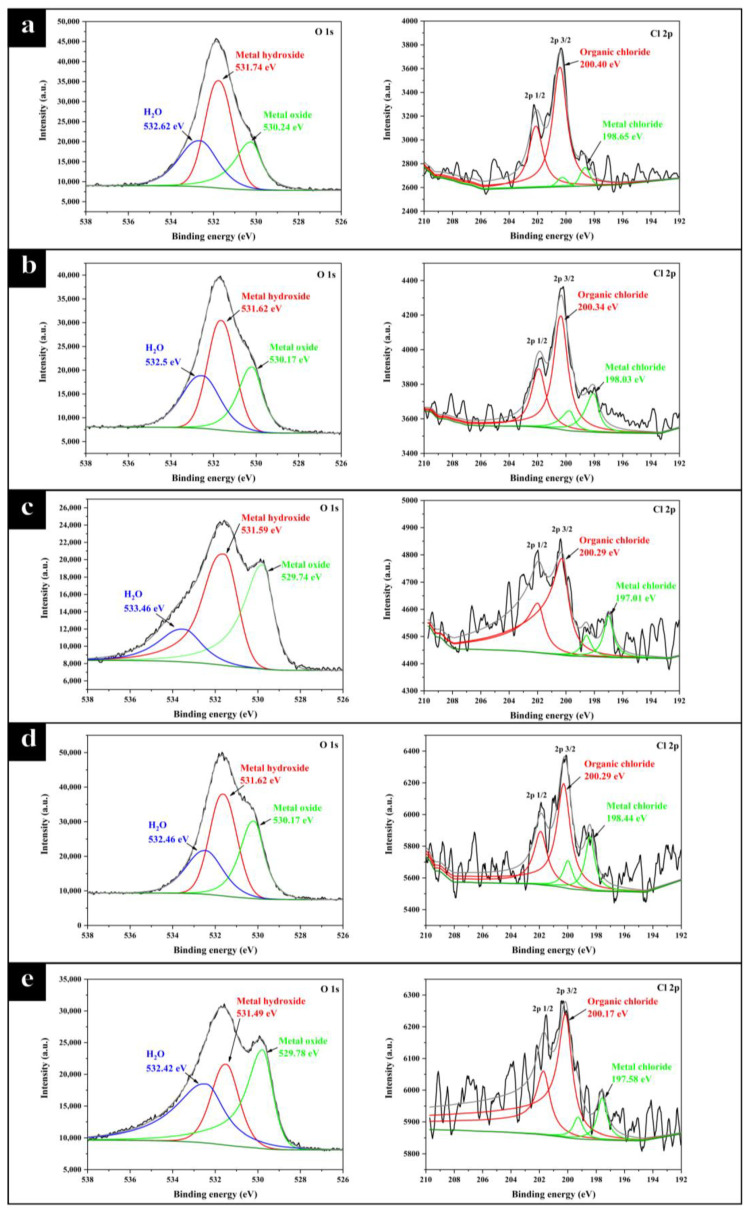
The XPS high-resolution spectrum with the signal of O 1s, Cl 2p (**a**) E4 group, (**b**) E4-CP1 group, (**c**) E4-CP2 group, (**d**) E4-CP4 group, and (**e**) E4-CP6 group.

**Table 1 molecules-28-03648-t001:** *E*_corr_ (V), *I*_corr_ (μA cm^−2^), and Tafel slope obtained from potentiodynamic polarization curves shown in Figure 1 (scan rate 1 mV s^−1^; 25 °C).

	*E*_corr_/V	*I*_corr_/μA cm^−2^	An Slope	Ca Slope
E4	−0.34	2.566	1.862	6.019
E4-CP1	−0.35	1.152	1.897	6.619
E4-CP2	−0.33	0.764	1.767	6.844
E4-CP4	−0.35	1.509	2.111	5.996
E4-CP6	−0.36	0.938	1.818	6.518
Water	−0.16	0.081	6.429	9.227

**Table 2 molecules-28-03648-t002:** The electrochemical impedance spectroscopy (EIS) fitting results with the relative error (frequency scan range from 10^6^ Hz to 1 Hz under the OCP).

	*R*s	CPE1		*R* _film_	CPE2dl		*R* _ct_
	Ω·cm^−2^	*T* (10^−9^ s^n^ Ω^−1^·cm^−2^)	*n*	Ω·cm^−2^	*T* (10^−6^ s^n^ Ω^−1^·cm^−2^)	*n*	Ω·cm^−2^
E4	36.07 (0.04)	2.99 (0.12)	1.00 (0.09)	248.6 (0.10)	3.48 (0.04)	0.83 (0.006)	4241 (0.02)
E4-CP1	38.27 (0.04)	3.19 (0.13)	0.99 (0.10)	259.5 (0.01)	1.02 (0.03)	0.89 (0.004)	8770 (0.02)
E4-CP2	35.71 (0.05)	2.99 (0.21)	1.00 (0.01)	169.4 (0.02)	0.89 (0.03)	0.87 (0.005)	9778 (0.02)
E4-CP4	33.96 (0.06)	2.38 (0.19)	1.00 (0.01)	194.4 (0.02)	3.30 (0.05)	0.76 (0.008)	5086 (0.03)
E4-CP6	37.51 (0.03)	4.43 (0.16)	1.00 (0.01)	135.9 (0.01)	0.93 (0.02)	0.89 (0.003)	8941 (0.02)

**Table 3 molecules-28-03648-t003:** The Binding energy (BE /eV), Full Width Half Maximum (FWHM /eV) and area of Fe 2p, Cr 2p, O 1s, Cl 2p in group E4, E4-CP1, E4-CP2, E4-CP4, and E4-CP6, respectively.

Components		E4		E4-CP1		E4-CP2		E4-CP4		E4-CP6	
	FWHM (eV)	BE (eV)	Area (a.u.)	BE (eV)	Area (a.u.)	BE (eV)	Area (a.u.)	BE (eV)	Area (a.u.)	BE (eV)	Area (a.u.)
Fe met	2	707.3	1591.68	707.3	1406.82	707	3301.96	707.2	2761.18	707	3393.58
FeO	1.27	709.4	2148.63	709.3	2666.73	709.2	2738.41	709.2	5130.01	709	3062.91
Fe_3_O_4_	0.97	710.3	1607.4	710.1	2885.96	710.1	1902.64	710.1	3546.17	710	3261.37
Fe_2_O_3_	1.14	710.8	2329.76	710.8	3166.89	710.8	2405.36	710.6	3069.4	710.7	3763.99
FeCl_3_	1.03	711.3	977.63	711.2	1039.56	711.3	653.1	711.3	1279.99	711.3	956.38
α-FeOOH	2.17	711.8	8382.02	711.8	9939.37	711.8	4205.05	711.4	12,402.6	711.8	7849.99
Fe_sat_Ⅱ	3.14	713.6	7066.94	713.6	3811.13	712.3	5596.48	712.7	8843.04	712.6	4843.91
Fe_sat_Ⅲ	3.5	716.6	7451.96	715.1	1624.68	714.1	4085.5	713.7	3446.05	714.3	3433.72
Cr_2_O_3_	1.33	576.6	4536.04	576.3	6260.67	576.4	1790.08	576.3	8183.33	576.7	1791.15
Cr(OH)_3_	1.45	577.4	3955.75	577.3	4196.74	577.4	1497.42	577.1	5832.49	577.4	2524.22
CrO_3_	2	578	2340.47	578.4	2025.91	578.6	2414.5	578.1	6101.37	578.3	3393.58
Metal Oxide	1.5	531.7	23,191.4	531.6	22,758.4	531.6	25,927.4	531.6	36,335.4	531.5	31,226.5
Metal hydroxides	1.15	530.2	43,355.4	530.2	37,254.4	529.7	29,460.4	530.2	47,145.4	529.8	21,768.8
Bounded water	2	532.6	28,109.9	532.5	26,676.5	533.5	10,670.2	532.5	29,327.7	532.4	32,464.4
Organic chloride^2p3^	1.13	200.4	1980	200.3	1273.23	200.3	1125.89	200.3	1437.22	200.2	1217.16
Organic chloride^2p1^	1.13	202.1	995.42	201.9	642.95	202	564.61	201.9	707.71	201.7	587.65
Metal chloride^2p3^	0.81	198.7	215.59	198	642.95	197	187.83	198.4	426.44	197.6	164.37
Metal chloride^2p1^	0.81	200.3	105.72	199.7	183.29	198.6	96.57	200	218.83	199.3	84.47

**Table 4 molecules-28-03648-t004:** The main elemental composition of 1Cr11Co3W3NiMoVNbNB stainless steel.

Elements	C	Mn	Cr	Mo	V	Ni	Co	W	Nb	Fe
wt%	0.08~0.13	0.35~0.65	10~12	0.10~0.40	0.15~0.25	0.3~0.7	2.5~3.5	2.4~3.0	0.05~0.12	Balance

## Data Availability

Not applicable.

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
