# Peer review of "Passivation Effect of the Chlorinated Paraffin Added in the Cutting Fluid on the Surface Corrosion Resistance of the Stainless Steel"

_molecules, 2023, doi:10.3390/molecules28093648_

Round 1

Reviewer 1 Report

The manuscript describes the investigation of the passivation effect of differing concentration chlorinated paraffin on the machined surface of stainless steel was studied, through electrochemical measurements and surface morphology investigation.

The topic is novel, the scope is narrow; the methods are limited and results are appropriate for achieving the objectives.

However, the reproducibility is relatively low with some experimental details omitted or not clearly introduced, compounded by the missing product information for each instrument used.

English usage needs to be significantly improved, even by a professional proof-reading service. Statistical analysis should have been done to facilitate comparison amongst the groups.

In conclusion, a major revision is needed before further consideration in Molecules.

More details for consideration:

1. The narrow scope is the result of the first sentence limiting it to “industrial fields,” where molecular scale is barely relevant. Please expand the scope by including biomedical field and cite the latest literature doi:10.4012/dmj.2016-206, 10.3390/nano9081119

2. Rewrite some poorly-written sections: e.g. L40-L41, L45-L54, L165… there are more.

3. “average surface roughness is lower than 100nm.” How was this determined?

4. “provided by Phenom ProX” is not sufficient product information. A complete product information should contain: serial or batch No., brand name, company, city and country.

5. 30min, a space is needed in between.

6. Equations should be numbered.

7. Fig. 6, please adjust the y-axis range to eliminate the large blank space above and beneath. Why are the error bars so big? This fig. is not very informative.

8. Fig. 7, what is (a)? The scale bars are barely visible, so are EDS results.

9. Conclusion, L429-441 is repeat of results and should be eliminated.

10. The statements “…is consistent with the literature [45],” and “…consistent with the EDS results, as shown in Fig. 7 (c),” both need to be elucidated.

Reviewer 2 Report

In this paper, the passivation effect of CP with different concentrations on the machined surface of stainless steel was studied. There are two major drawbacks to this work. Firstly, there had no in-depth understanding and discussion for the corrosion mechanism. Secondly, although some results were obtained, the experimental results are far from sufficient to support the conclusion. Some specific points are as follows:

1.     The Abstract is tedious and should be simplified.

2.     For the potentiodynamic polarization curves:

1)      How are the values of the electrochemical parameters Icorr obtained from the potentiodynamic polarization curves? The polarization curves of the alloy are typically passive. In this case, the Tafel extrapolation method is not formally valid to determine Icorr.

2)      “The Icorr value continuously decreased with the addition of 1mL, 2 mL, 4 mL and 6 mL CP, respectively.” This conclusion is wrong. This is a very serious problem for this research work.

3)      The purpose of the authors in this study is to clarify the effect of the chlorinated paraffin on the passive behavior. Thus, it is more important to study the passive current density instead of Icorr.

4)      “The value of Epp depended on the oxidation of Cr to Cr3+” This is unconvincing because the Epp is great larger than the potential for the oxidation of Cr to Cr3+.

3.     For the EIS results:

1)      Why EIS tests were performed after the polarization tests.

2)      Since the EIS tests were performed after the polarization tests, thus there is no direct correlation between the EIS results and the Icorr. So some conclusion, such as “A smaller semicircle diameter means worse corrosion resistance which showed a good agreement with the corrosion current density Icorr obtained from the polarization curves” are unreasonable.

3)      The test frequency, from 106 Hz to 1 Hz, is confusing. It may be more reasonable from 106 Hz to 10-2 Hz.

4)      The equivalent circuit is puzzling. From Fig. 5, the Rfilm should be the resistance of the corrosion scale and the Rct should be the resistance of passive film.

5)      “The CPE value is relevant to the thickness of the passivation film. The increase of CPE is caused by the decrease of the thickness of the passivation film” This is inadequate.

6)      “The Rct values increased with the increase of CP concentration” This conclusion is wrong.

4.     For the SEM results:

1)      Fig. 7 should be modified because some annotation is too small to see straight.

2)      The highlighted part in the Fig. 7a is second phase?

5.     For the XPS results:

1)      How to perform the XPS tests? As shown in Fig. 5, there has a layer of corrosion scale on the passive film. How can the authors confiem that passive films instead of corrosion scale was tested.

2)      There has some problem for the fitting of the high-resolution spectra. For example, it is not recognized for the fitting of Fe 2p spectra because the difference of the binding energy for some peaks is too small to distinguish. In addition, the error for some fitting is too large, such as 578 eV and 578.64 eV for the same CrO3.

6.     There is no depth discussion on the mechanisms of corrosion resistance.

7.     There are a number of irregularities in the writing of the paper, such as the italic, should be check and modify.

Round 2

Reviewer 1 Report

The issues have been addressed. It it ready for publication.

L394, the charge of Me should be n+